# Highly Expressed Progesterone Receptor B Isoform Increases Platinum Sensitivity and Survival of Ovarian High-Grade Serous Carcinoma

**DOI:** 10.3390/cancers13215578

**Published:** 2021-11-08

**Authors:** Hao Lin, Kuo-Chung Lan, Yu-Che Ou, Chen-Hsuan Wu, Hong-Yo Kang, I-Chieh Chuang, Hung-Chun Fu

**Affiliations:** 1Department of Obstetrics and Gynecology, Kaohsiung Chang Gung Memorial Hospital and Chang Gung University College of Medicine, Kaohsiung 83341, Taiwan; haolin@cgmh.org.tw (H.L.); blue@cgmh.org.tw (K.-C.L.); tedycou@gmail.com (Y.-C.O.); chenhsuan@cgmh.org.tw (C.-H.W.); hkang3@cgmh.org.tw (H.-Y.K.); 2Center for Menopause and Reproductive Medicine Research, Kaohsiung Chang Gung Memorial Hospital and Chang Gung University College of Medicine, Kaohsiung 83341, Taiwan; 3Department of Obstetrics and Gynecology, Jen-Ai Hospital, Taichung 41265, Taiwan; 4Department of Obstetrics and Gynecology, Chia-Yi Chang Gung Memorial Hospital, Chia-Yi 61363, Taiwan; 5Graduate Institute of Clinical Medical Sciences, Chang Gung University, Lin-Kou 33302, Taiwan; 6Department of Anatomic Pathology, Kaohsiung Chang Gung Memorial Hospital and Chang Gung University College of Medicine, Kaohsiung 83341, Taiwan; b9205043@cgmh.org.tw

**Keywords:** progesterone receptor-B, platinum sensitivity, ovarian high-grade serous carcinoma, prognosis

## Abstract

**Simple Summary:**

Ovarian high-grade serous carcinoma is the deadliest ovarian cancer. Cancer cells develop resistance to anti-cancer regimens leading to poor prognosis. Previous studies showed that the progesterone receptor was associated with better rates of survival of ovarian cancer patients. We aimed to investigate the association between the progesterone receptor and its isoform-B and platinum sensitivity of ovarian high-grade serous carcinoma. We found that strong progesterone receptor-B expression was associated with better platinum sensitivity and better survival in high-grade serous ovarian cancer patients. Our clinical data also showed that a high expression of progesterone receptor-B and optimal debulking were the independent factors associated with better platinum sensitivity. In a cell model, enhancing progesterone receptor-B expression and progesterone treatment increased platinum sensitivity and platinum-related apoptosis of the ovarian cancer cells. These might be potential therapeutic targets of ovarian high-grade serous carcinoma.

**Abstract:**

Background: Expression of the progesterone receptor (PR) has been reported to influence survival outcomes in patients with ovarian high-grade serous carcinoma (HGSC). In the present study, we attempted to investigate the association among PR and its isoforms’ expression, platinum sensitivity, and survival in ovarian HGSC. Material and methods: This retrospective study reviewed ovarian HGSC patients who received surgery followed by adjuvant chemotherapy. We analyzed total PR and PR isoform-B (PR-B) expression by immunohistochemical staining and quantified using the H-score. Then, we compared platinum sensitivity and survival outcomes between those patients with weak and strong PR-B expression. Cisplatin viability assays were carried out in ovarian HGSC cell lines (OC-3-VGH and OVCAR-3) with different PR-B expression. Results: Among 90 patients, 49 and 41 patients were considered to have platinum-sensitive and platinum-resistant disease, respectively. Pearson’s correlation model showed that the H-score of total PR correlated positively with PR-B (*r* = 0.813). The PR-B H-score of tumors was significantly higher in the platinum-sensitive group (*p* = 0.004). Multivariate analysis revealed that the PR-B H-score and optimal debulking status were independent factors predicting platinum sensitivity. When compared with strong PR-B expression, patients with weak PR-B had significantly poorer progression-free (*p =* 0.021) and cancer-specific survival (*p* = 0.046). In a cell model, cisplatin-resistant OC-3-VGH cells expressed a lower level of PR-B than wild-type cells. Overexpression of PR-B or progesterone could increase cisplatin sensitivity in both OC-3-VGH and OVCAR-3 cells via the mechanism of promoting cisplatin-related apoptosis. Conclusions: When compared to weak PR-B, ovarian HGSC patients with a strong PR-B expression had a better chance of platinum sensitivity and survival, and this finding was compatible with our experimental results. Progesterone seemed to be a platinum sensitizer, but the value of adding progesterone in the treatment of ovarian HGSC should be further investigated.

## 1. Introduction

The standard treatment for advanced-stage epithelial ovarian cancer (EOC) is primary debulking surgery followed by adjuvant chemotherapy. Although most patients respond to platinum-based chemotherapy initially, relapses are common, leading to platinum resistance, which is uniformly fatal. Recent revolutions in cancer treatments from targeted therapies to immunotherapies have provided hope in treating cancer patients, but some women with EOC have not benefited much from these advancements. The major reason is that they can easily develop resistance to these powerful anti-cancer regimens leading to poor prognosis and survival [1,2]. Therefore, novel therapeutic approaches for overcoming drug resistance are emerging.

Hormone therapy is occasionally used as a treatment option in patients with relapsed ovarian cancer who have finally exhausted multiple lines of systemic chemotherapy. Although there is a strong evidence base to support the use of hormonal therapy in the treatment of hormone-dependent breast cancer and endometrial cancer [3,4], there are few prospective clinical trials evaluating this approach in recurrent ovarian cancer [5,6]. Pre-clinical models have demonstrated that ovarian cancer cells that express estrogen receptor (ER) could be inhibited by anti-estrogens, providing the rationale for the use of tamoxifen, letrozole, or anastrozole in this disease [7]. Previous studies demonstrated that between 0 and 15% of response rates depend on receptor status [8,9]. Epidemiological evidence strongly suggests progesterone-containing contraceptives and pregnancy appear to provide a protective effect for EOC occurrence, implying the possible role of progesterone receptor (PR) in cancer development [10]. A recent meta-analysis has shown that strong PR expression was independently associated with improved disease-specific survival in patients with ovarian cancer [11]. Rodriguez et al. found that progesterone treatment of ovarian cancer cells in vitro resulted in decreased proliferation and increased apoptosis [12].

Among ovarian cancers, ovarian high-grade serous carcinoma (HGSC) is the deadliest and common ovarian cancer [13]. Sieh W. et al. showed that the proportion of tumors that stained positive (≥1%) for PR was 31.1% for ovarian HGSC. Additionally, PR expression could be associated with better survival of ovarian HGSC [14]. Le Page C. et al. reported diffuse PR expression associated with a survival benefit of advanced ovarian HGSC, but not in those with early-stage diseases [15]. However, some investigators reported that PR expression was not associated with the survival benefit of ovarian HGSCs [16,17].

Human PR has two major isoforms A (PR-A) and B (PR-B), and differences in the expression are often observed in many hormone-dependent malignancies. In breast cancer, PR-A predominant tumors are more sensitive to hormonal treatment, while PR-B predominant tumors are more common in advanced stage patients with poor prognosis [18]. In endometrial cancer, both PR-A and PR-B are related to low-risk disease; however, only PR-B predicts a better prognostic outcome [19]. The role of each receptor isoform remains unclear with regards to chemotherapy responsiveness and prognosis of ovarian cancers.

However, at this point in time, detailed mechanistic studies are lacking, and the correlation between PR expression, survival, and chemosensitivity in ovarian HGSC is not well known. Thus, in present study, we attempted to investigate the association between PR and its isoform B expression, chemosensitivity, and survival in ovarian HGSC patients and cell-line models.

## 2. Material and Methods

### 2.1. Patients Enrolled

This study is a retrospective design. We reviewed patients with ovarian HGSC between January 2010 and December 2015 in Kaohsiung Chang Gung Memorial hospital. The stages were defined according to the International Federation of Gynecology and Obstetrics (FIGO) 2014 system [20]. Patients were included if they underwent debulking surgery followed by more than 4 courses of adjuvant chemotherapy, aged between 20 to 80, with regular follow-up and sufficient tissues for immunohistochemical (IHC) staining. Patients with double cancers were excluded. Clinical data including age, optimal debulking or not, FIGO staging, carcinoembryonic antigen (CEA) and carbohydrate antigen-125 (CA-125) levels, platinum-sensitivity, and survival time were collected from the medical records. Platinum-sensitivity was assigned to resistant or sensitive according to the time relapsed (6 months as cut-off) since completing first-line chemotherapy [21]. Clinical variables, platinum-sensitivity, and survival outcomes were compared between weak and strong PR-B IHC expression groups. The Institutional Review Board of Chang Gung Memorial Hospital (approval number: 202000565B0) has reviewed and approved this study protocols.

### 2.2. Immunohistochemical Staining Analysis and Scoring for PR

Formalin-fixed, paraffin-embedded tissue sections were prepared to evaluate the expressions of the PR. The slides were deparaffinized with a xylene rinse and rehydrated with indicated concentrations of alcohol (100%, 95%, 85%, and 75%). Then, we rinsed them with distilled water. Antigen retrieval was performed with heat-mediated method by citrate buffer (10 mM, pH 6.0). We used 3% hydrogen peroxide solution to quench endogenous peroxidase activity. The primary PR antibodies (1:100, Leica, US. Cat# PR NCL-L-PGR-312, for both the PR-A and PR-B of PR; 1:100, Merck, USA., Cat# MABS1234, for PR-B only) were incubated. The antigen–antibody complexes were detected with a secondary antibody with peroxidase-conjugated streptavidin. Peroxidase activity was demonstrated with diaminobenzidine (DAB) (Dako, Glostrup, Denmark) and lightly counterstained with Gill’s hematoxylin (Merck, Whitehouse, NJ, USA). Expression PR/PR-B staining was scored using the histoscore (H-score). The score was obtained by multiplying the nuclei cell intensity (graded as 0: no stained, 1: weak, 2: moderate, 3: strong) by the percentage of positive cells (on a scale of 0 to 100). The range of H-score was 0 to 300 [22]. The pathologist (Chuang IC) blindly scored all the cases.

### 2.3. Cell Culture and Generation of Cisplatin-Resistant Ovarian Cancer Cell Lines

Ovarian high-grade serous cancer cell lines, OC-3-VGH and OVCAR-3 cells, were obtained from Bioresource Collection and Research Center (BCRC, Hsinchu, Taiwan) and cell line authentication was performed by BCRC. OC-3-VGH cells were cultured in Dulbecco’s Modified Eagle Medium: Nutrient Mixture F-12 (DMEM/F12) with 1.5 g/L sodium bicarbonate and 10% fetal bovine serum, 100 IU/mL penicillin, 100 mg/mL streptomycin, and 0.4 mM L-glutamine (Sigma, St. Louis, MO, USA). OVCAR-3 cells were cultured in Roswell Park Memorial Institute (RPMI, Buffalo, NY, USA) 1640 with 1.5 g/L sodium bicarbonate, 2 mM L-glutamine, 2.5 g/L glucose, 10 mM HEPES, 1 mM sodium pyruvate, 10 ng/mL insulin and 20% fetal bovine serum, 100 IU/mL penicillin, 100 mg/mL streptomycin, and 0.4 mM L-glutamine (Sigma). Both cell lines were cultured in a humidified 95% atmosphere with 5% CO_2_ at 37 °C. We developed acquired cisplatin-resistant OV3-VGH cells using a stepwise increase in treatment concentrations (1 to 6 μM) with cisplatin (Sigma, St. Louis, MO, USA, Cat# 479306). This development period was carried out for about 2 months. Finally, the cisplatin-resistant cells (OC-3-VGH-resist) were kept in the DMEM/F12 medium with 6 μm cisplatin.

### 2.4. Transfection of Ovarian Cancer Cell Line

The plasmids of pcDNA3-PRB were acquired through Addgene [23]. We transfected cells using Lipofectamine 2000 reagent (Life Technologies, Carlsbad, CA, USA) according to the manufacturer’s instructions. Stable PR-B overexpressing cell lines (OC-3-VGH-PR-B and OVCAR-3-PR-B) were generated using pcDNA3-PRB harboring wild type PR-B. Cells transfected with the empty vector pcDNA3 were used as controls (OC-3-VGH-vector and OVCAR-3-vector). All transfection experiments were confirmed via real-time polymerase chain reaction and Western blotting. After transfection, the cells were grown in Neomycin (G418, Sigma) at 200 ug/mL for stable clone selection.

### 2.5. Western Blotting

We lysed the cells with lysis buffer with 1% protease inhibitor cocktail (Roche Applied Science, Indianapolis, IN, USA). Proteins were extracted from whole cells, resolved by sodium dodecyl sulfate-polyacrylamide gel electrophoresis and transferred to polyvinylidene fluoride membranes. Membranes were probed with anti-PR-B (1:500, Merck, Cat#MABS1234) and anti-actin (1:10,000; Millipore, Burlington, MA, USA, Cat#MAB1501). Then, membranes were incubated with horseradish peroxidase-conjugated secondary antibody (Amersham, Buckinghamshire, UK). Enhanced chemiluminescence was read with ECL-Plus (Amersham Pharmacia, Hong Kong). The images were read by a LAS-3000 imager system (Fujifilm, Tokyo, Japan). Densitometric quantification of the bends was performed using Bio-Rad Quantity One 1-D Analysis software. Relative levels of PR-B expression were determined by normalization to the expression of β-actin.

### 2.6. Quantitative Real-Time Polymerase Chain Reaction

Quantitative real-time polymerase chain reaction (RT-PCR) was performed using Fast SYBR^TM^ Green Master Mix (Thermo Fisher, Waltham, MA, USA, Cat# 4385612) and Applied Biosystems^®^ 7500 Fast Real-Time PCR Systems. The cycling parameters for all genes were the following: hot-start 95 °C 15 min, 40 cycles of (denaturation 95 °C 3 s, annealing 60 °C 30 s, elongation 72 °C 30 s, plate read). SPSS software was used to analyze results. Primer sets: PR-B (forward: 5-ACTGAGCTGAAGGCAAAGGGT-3 and reverse: 5-GTCCTGTCCCTGGCAGGGC-3) and β-ACTIN (forward: 5-TCACCCACACTGTGCCCATCTACG-3 and reverse: 5-CAGCGGAACCGCTCATTGCCAATG-3).

### 2.7. Cellular Toxicity via CCK-8 Assay

The cellular viability of the OC-3-VGH and OVCAR-3 cells after incubation with cisplatin of different concentrations was determined by a cell counting kit-8 (CCK-8, Dojindo Molecular Technologies, Rockville, MD, USA) as a cellular toxicity assay. The OC-3-VGH and OVCAR-3 cells were seeded on the 96-well plate at the density of 3000 cells. Then, the cells were treated with indicated concentrations of cisplatin (0–6 μM for OC-3-VGH cells, 0–40 μM for OVCAR-3 cells), progesterone (P4, TargetMol, Boston, MA, USA, #T0478), or vehicle (Phosphate- Buffered Saline) for 24 h. We added CCK-8 and used enzyme-linked immunosorbent assay reader (Wallac Victor2 V, PerkinElmer, Waltham, MA, USA) to detect the absorbance (Optical Density, OD) at 450 nm.

### 2.8. Apoptosis Assay

The apoptosis assay of the OC-3-VGH-vector and OC-3-VGH-PR-B cells/OVCAR-3-vector and OVCAR-3-PR-B cells was determined using an FITC Annexin V Apoptosis Detection Kit I (BD, Cat#556547). The assay was performed according to the manufacturer’s protocol. In brief, the cells were treated with cisplatin (6 μM) for 24 h and 48 h, which was collected and subjected to the analysis. The cells were analyzed by flow cytometry (BD LSR II; BD Biosciences, San Jose, CA, USA). A minimum of 10,000 cells were then analyzed. All tests were performed in triplicates and repeated three times.

### 2.9. Statistical Analysis

We used Pearson correlation to find the correlation between the total PR and PR-B H-score. We identified the optimal cut-off values of the PR-B H-score for predicting platinum sensitivity using receiver operating characteristic (ROC) curves and the Youden index [24]. We used stepwise multivariate logistic regression analysis to determine independent factors predicting platinum sensitivity. The continuous data were checked with the Kolmogorov–Smirnov normality test. For comparisons of mean values, Student’s *t*-test was performed for data with normal distribution (age) and a non-parametric test was applied to analyze data without normal distribution (CEA, CA-125, treat-free interval and PR-B score). Chi-square test was used to compare frequency distributions between categorical variables. Progression-free survival (PFS) and cancer-specific survival (CSS) were defined as the interval from the date of diagnosis to the date of first evidence of recurrence and disease-specific death, respectively. Actuarial rates of PFS and CSS were estimated with the Kaplan–Meier method, and statistical differences between groups were determined using the log-rank test. Multivariate Cox proportional hazards analysis was used to identify the significant prognostic factors for PFS and CSS.

The Student’s *t*-test was used to analyze statistical significance between control and treatment groups (RT-PCR and Western blotting). The 2-way ANOVA with Bonferroni post hoc test was used to assess differences of viability between control and experimental groups (CCK-8 assay) treated with cisplatin. Statistical analysis was performed using SPSS package version 22 (IBM, Armonk, NY, USA) for Mac. A *p* value < 0.05 was taken to indicate statistical significance.

## 3. Results

### 3.1. Clinical Data

During the study interval, a total of 376 ovarian cancer patients were identified. We excluded 247 patients for non-HGSC histology, 26 patients for chemotherapy of less than 4 courses, and 13 patients for inadequate tissue for IHC staining or double cancer. Eventually, 90 patients met inclusion criteria and were enrolled for this study cohort. According to the ROC curve analysis, we found that the optimal cut-off value of PR-B H-score to predict platinum-sensitivity was 12.5 (AUC 0.664, 95% CI: 0.552–0.776). An H-score of 12.5 or higher was assigned as strong PR-B expression while lower than 12.5 was assigned as weak PR-B expression. Figure 1A–C demonstrate typical examples of IHC staining of PR. A significant positive correlation between total PR and PR-B H-score was found (*r* = 0.831; *p* < 0.001, Figure 1D). Among 90 patients, 49 and 41 patients were defined as having platinum-sensitive and platinum-resistant disease, respectively. Platinum-sensitive patients had a significantly higher mean PR-B H-score than those with platinum-resistant disease (29.4 ± 7.4 vs.10.4 ± 5.2, *p* = 0.005; Figure 1E).

Table 1 shows the comparison of patients’ characteristics between platinum-sensitive and platinum-resistant groups. Besides from higher PR-B H-score, platinum-sensitive patients had also more optimal debulking status and less advanced stage disease. Stepwise multivariate logistic regression analysis revealed that the PR-B H-score (OR 3.69; 95% CI 1.18–11.6) and optimal debulking status (OR 5.56; 95% CI 2.09–14.8) were the only independent factors predicting platinum resistance (Table 2).

The five-year PFS rate was significantly better in the strong PR-B group (46.2%) than the weak group (23.8%) (*p =* 0.021); the five-year CSS was also better in the strong PR-B group (71.6%) than the weak PR-B group (47.7%) (*p* = 0.046) (Figure 2). In multivariate Cox regression analysis, optimal debulking status was the only significant independent factor associated with both PFS and CSS, while PR-B was marginal significance only for PFS (Appendix A).

### 3.2. Western Blot Analysis and Quantitative Real-Time Polymerase Chain Reaction on Ovarian HGSC Cells with Different PR-B Expression

OVCAR-3 cells had less PR-B expression than OC-3-VGH cells both in protein and mRNA levels (OC-3-VGH vs. OVCAR-3: 1 ± 0.07 vs. 0.39 ± 0.02) (Figure 3A). Relatively, OVCAR-3 cells were more resistant to cisplatin treatment than OV-3-VGH cells (Figure 3A). OC-3-VGH and OVCAR-3 cells were transfected with the control vector (OC-3-VGH-vector and OVCAR-3-vector) or plasmids pcDNA3-PR-B (OC-3-VGH-PR-B and OVCAR-3-PR-B). After transfection, we picked up stable clones with G418 200 ug/mL. Protein was extracted and subjected to Western blot. Immunoblotting showed that about 100% more PR-B protein expression in OC-3-VGH-PR-B when compared to OC-3-VGH-vector cells (Figure 3B). OC-3-VGH-PR-B cells also demonstrated significant higher level of PR-B mRNA (OC3-VGH-vector vs. OC3-VGH-PR-B: 1 + 0.03 vs. 2.47 ± 0.05) (Figure 3B). In OVCAR-3 cells, there was about 170% more PR-B protein and 110% PR-B mRNA expression in OVCAR-3-PR-B than OVCAR-3-vector cells (OVCAR-3-vector vs. OVCAR-3-PR-B: 1 ± 0.02 vs. 2.12 ± 0.03) (Figure 3C).

### 3.3. PR-B Expression Promotes Cytotoxic Effects of Cisplatin on Ovarian HGSC Cell

We checked the cells’ viability with various concentrations of cisplatin (0–6 μM for OC-3-VGH cells; 0–40 μM for OVCAR-3 cells) for 24 h. To determine the effects of PR-B expression on the cytotoxic effects of cisplatin, we found that ovarian HGSC cells with a higher expression of PR-B were more sensitive to platinum treatment (OV3-VGH, *p* = 0.003; OC-3-VGH-PR-B, *p* = 0.002 and OVCAR-3-PR-B, *p* = 0.001) than control cells (Figure 3B,C).

### 3.4. Effects of Progesterone Treatment on Acquired Cisplatin-Resistant Ovarian HGSC Cells

We established an acquired cisplatin-resistant cell line (OC-3-VGH-resist) from the parental OC-3-VGH cell line by incubating cells with increasing concentrations of cisplatin. The cells were kept in the medium with 6 μM cisplatin. OC-3-VGH-resistant cells expressed significantly lower levels of PR-B protein and mRNA (1 ± 0.02 vs. 0.78 ± 0.05) than parental OC-3-VGH cells (Figure 4A,B). OC-3-VGH-resistant cells were more resistant to cisplatin than parental OC-3-VGH cells. Cell viability was then assessed in different conditions. Compared to single cisplatin treatment, the combination of P4 with cisplatin reduced the viability of OC-3-VGH-resistant cells. Additionally, transient overexpression of PR-B in OC-3-VGH-resistant cells (OC-3-VGH-resist-PR-B) also reduced viability of the cells in various concentrations of cisplatin (Figure 4C). The data showed that PR-B protein and P4 re-sensitized the OC-3-VGH-resistant cells to cisplatin treatment.

### 3.5. Effects of Progesterone Treatment on Ovarian HGSC Cell Survival with Different PR-B Expression

Ovarian HGSC cells with different PR-B expression were incubated with various concentrations of cisplatin and different concentrations of P4 for 24 h. The cell viability was then assessed using a cell counting kit-8. We found that P4 (0.1 μM) synergistically promoted the inhibition of viability by cisplatin in all types of ovarian HGSC cells. However, further inhibition of cell viability was not observed if P4 concentrations were adjusted to 1 and 10 μM (Figure 5).

### 3.6. Apoptosis of Ovarian HGSC Cells during Treatment of Cisplatin

To determine the mechanism for synergistic effects between PR-B expression and cisplatin treatment in inhibiting cell viability, we performed apoptosis assay with FITC-Annexin V and PI double staining. Flow cytometry analysis showed that about 59% of OC-3-VGH-PR-B cells and 50% of OVCAR-3-PR-B cells underwent apoptosis after treatment of cisplatin for 48 and 24 h, respectively. The percentage was significantly higher than control cells (Figure 6A,B).

## 4. Discussion

Our clinical data demonstrated that tumor with weak PR-B expression is associated with more platinum resistance and poor PFS in ovarian HGSC patients. Preclinical evaluation further showed that HGSC cells with weaker expression of PR-B are more resistant to cisplatin treatment and progesterone treatment could re-sensitize platinum-resistant ovarian HGSC cells to cisplatin. Novel therapeutic approaches involving regulation of PR might be promising for treatment of ovarian serous carcinoma patients.

In ovarian cancer, the association between PR and platinum sensitivity is seldom investigated. Epidemiological evidence suggests protective effect of progesterone for EOC occurrence [10]. In earlier reports, Tkalia et al. demonstrated that 63.4% of ovarian serous tumor specimens stained positive for PR [25]. However, in another study, PR immunopositivity was observed in only 8.9% of ovarian HGSC tissues [26]. In a more recent robust study investigating hormone-receptor expression and ovarian cancer survival in 2933 patients, the proportion of tumors that stained positive for PR was highest for endometrioid type (67.4%) and low-grade serous type (57.4%), followed by HGSC (31.1%), and lowest for mucinous type (16.4%) and clear cell type (8.0%) [14]. Our results demonstrated a positive staining rate of 26.8% for HGSC; the differences among various studies may be explained by different IHC antibodies and interpretation criteria used. In this robust study, PR expression was found to be strongly associated with better HGSC and endometrioid carcinoma survival. Further evaluation showed that strong PR expression was independently associated with significantly improved HGSC survival compared with PR-negative tumors after adjusting for site, age, stage, and grade [14]. The finding that strong PR expression was associated with improved HGSC survival is consistent with a recent meta-analysis combining all ovarian cancers, among which HGSC is the predominant subtype [11]. However, the treatment data were limited in these study cohorts, precluding an evaluation the association between hormone receptor status and chemosensitivity. In one preclinical study using OVCAR-3 cells, Peluso et al. showed that high PR expression was associated with decreased PR membrane component-1 expression, which in turn increased the effectiveness of cisplatin [27]. In the present study, although with a limited number of cases, we are the first to find that highly PR-B expressed HGSC patients were more sensitive to platinum-based chemotherapy, which translates to better PFS and CSS. This clinical finding is supported by our preclinical experiments that PR-B expressed ovarian cancer cells were more sensitive to cisplatin treatment. Similarly, platinum-sensitive ovarian cancer cells significantly harbored more PR-B both in mRNA and protein levels. Although several in vitro studies from over 20 years ago demonstrated an inhibitory action of P4 on ovarian cancer cell growth [28,29,30,31], one recent study found P4 protected ovarian cancer cells from cisplatin-induced cell cycle arrest and restored the cell migratory capability following treatment of cisplatin, indicating biphasic abilities of P4 to either induce or inhibit apoptosis [32]. In our study by adding P4, we found both ovarian cancer cells with weak and strong PR-B became more sensitive to cisplatin treatment. We believe that P4 may be useful as an adjunct to cisplatin therapy in ovarian HGSC patients regardless of PR-B status.

Until now, there have only been a few clinical studies on the use of P4 in the treatment of EOC. Chen and Feng reported in 2003 that P4 combined with platinum-based chemotherapy as a first-line therapy may improve the prognosis of advanced but not early stage EOC patients [33]. The results suggested that P4 could be used as an adjunct to platinum-based therapy. However, subgroup analysis with different hormone receptors status or histologic type had not been performed. The limited number of the clinical trial showing beneficial effects of P4 treatment may be due to its “double-edged sword effect” of either inhibiting or protecting ovarian cancer cell.

Human PR has two isoforms, A and B, which differ only in that the smaller PR-A lacks the N-terminal 164 amino acids of the larger PR-B. The nuclear receptors play different roles in the cells. PR-A and PR-B affect the target genes in a different manner. Generally speaking, PR-A acts as an inhibitor of PR-B [34]. An interesting finding in our study was that the total PR expression positively correlated with PR-B expression indicating a possible constant ratio of PR-A to PR-B in our patients. Currently, most commercially available antibody clones recognize epitopes located in the common region of PR to check both PR-A and PR-B. Some antibody clones recognize epitope on the N-terminal of PR-B, so they specifically detect PR-B only. Therefore, it is theoretically unable to discern PR-A from PR-B using immunohistochemistry techniques, although some studies claimed their ability to do so [35]. Ongoing studies are needed of using Western blotting to discriminate between the isoforms supporting if cisplatin is better option for tumors with a high PR-B to PR-A ratio.

This study has some limitations including its retrospective nature, missing some clinical data, potential patient-selection bias, a limited number of cases evaluated, only PR-B and total PR were investigated, heterogeneous therapies after recurrence, and varying follow-up practice patterns. Moreover, our preclinical experiments still need to be confirmed by either using primary cell cultures that closely resemble the physiology of cells in vivo or establishing a xenograft model in mice for animal study.

## 5. Conclusions

In conclusion, we demonstrated that tumors with weak PR-B expression were associated with more platinum-resistant and poor survival in ovarian HGSC patients. In a cell model, P4 and PR-B expression sensitize ovarian high-grade serous carcinoma cells to cisplatin via promotion of cisplatin-related apoptosis. Moreover, our preclinical evaluation further showed that P4 treatment could re-sensitize platinum-resistant ovarian HGSC cells to cisplatin, indicating that this approach has the potential to be used clinically to improve the efficacy of cisplatin therapy.

## Figures and Tables

**Figure 1 cancers-13-05578-f001:**
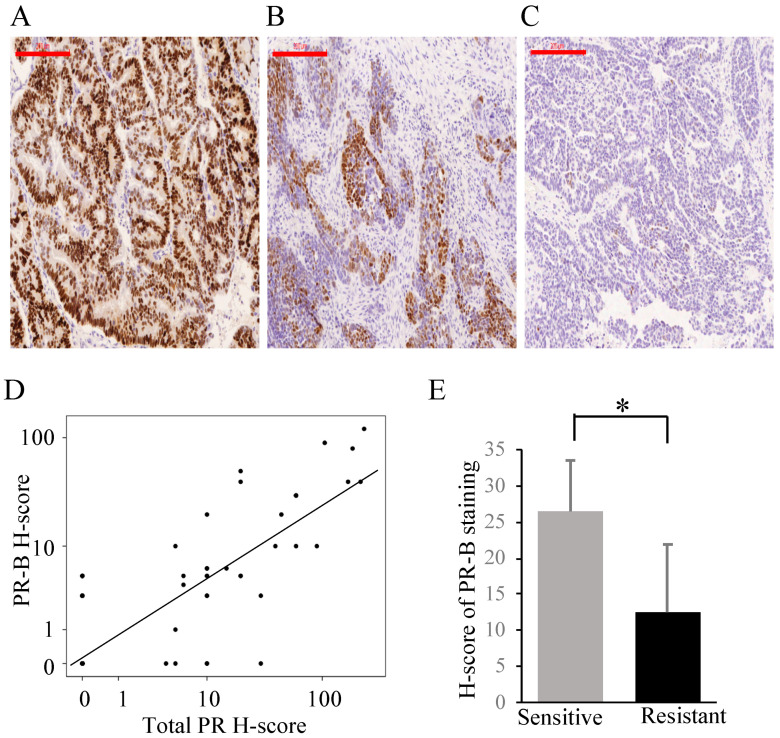
Progesterone receptor (PR) expression in platinum-sensitive and -resistant ovarian high-grade serous carcinoma (HGSC) tumors. (**A**–**C**) Representative images for PR-B immunohistochemical staining (scale bar = 200 μm). The intensities were strong (**A**), moderate (**B**) and weak(**C**). The percentages of positive cells were 70% (**A**), 20% (**B**), and 3% (**C**). H-scores of (**A**–**C**) were 210, 40, and 3, respectively. (**D**) The scatter plot (log-log scale) shows the positive correlation between total PR and PR-B H-score; Pearson’s correlation coefficient was 0.831, *p* < 0.001. (**E**) Data were expressed as mean PR H-score; platinum-sensitive patients had significantly higher H-score than platinum-resistant patients, *p* = 0.005. *: *p* < 0.05.

**Figure 2 cancers-13-05578-f002:**
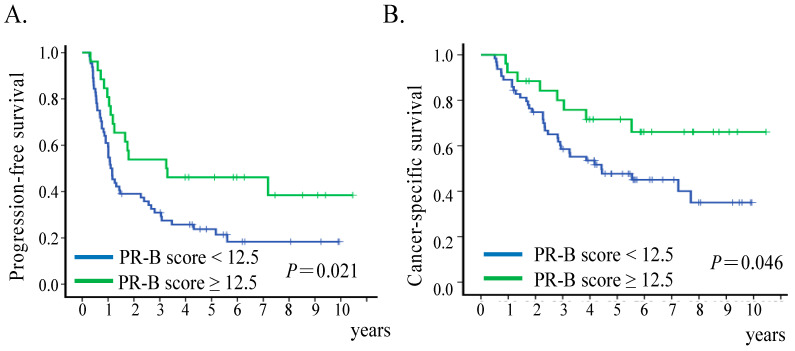
Survival outcomes of patients with different expressions of PR. Kaplan–Meier curves of progression-free survival (**A**) and cancer-specific survival (**B**) for patients who had weak PR-B expression (H-score < 12.5) or strong PR-B expression (H-score ≥ 12.5); *p*-values correspond to log-rank test.

**Figure 3 cancers-13-05578-f003:**
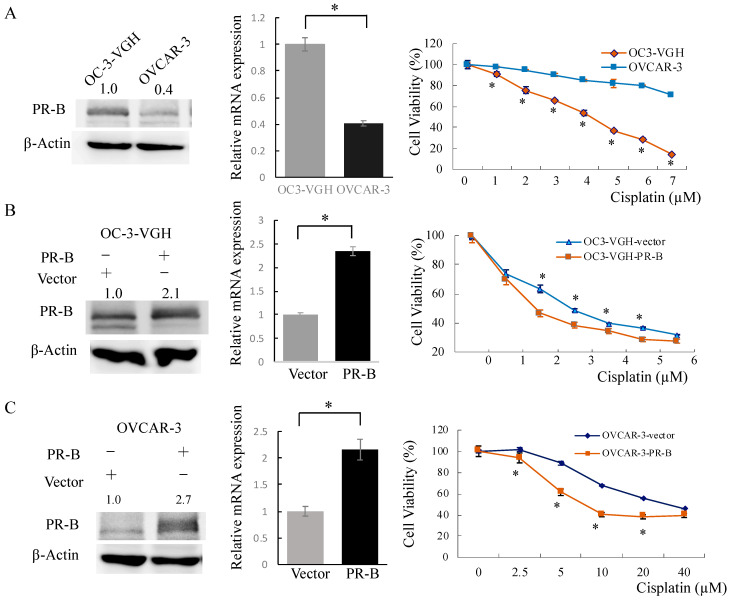
PR-B expression sensitizes ovarian high-grade serous cells to cisplatin treatment. (**A**) OVCAR-3 cells show lower PR-B expression at both the protein and mRNA levels and are more resistant to cisplatin treatment than OC-3-VGH cells. (**B**, **C**). OC-3-VGH-PR-B/OVCAR-3-PR-B cells show higher PR-B expression at both the protein and mRNA levels and are more sensitive to cisplatin treatment than OC-3-VGH-vector/OVCAR-3-vector cells. *: *p* < 0.05.

**Figure 4 cancers-13-05578-f004:**
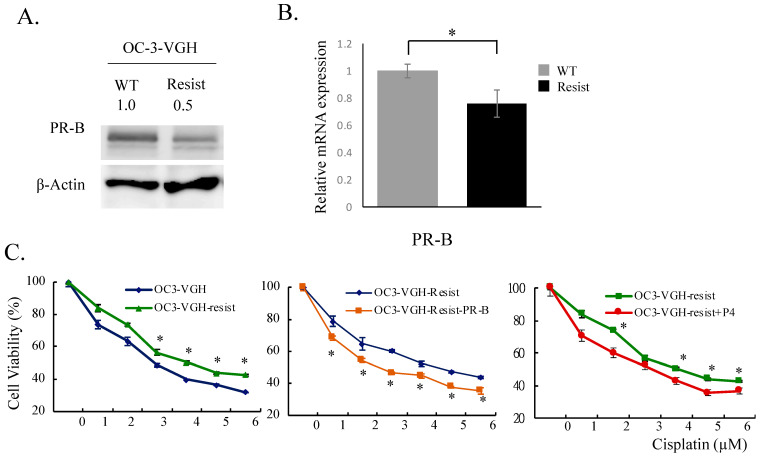
PR-B expression and progesterone (P4) re-sensitize platinum-resistant ovarian HGSC cells to cisplatin treatment. When compared to wild type (WT) OC-3-VGH cells, platinum-resistant cells express significantly lower PR-B both in protein (**A**) and mRNA (**B**) levels. P4 treatment or acquired PR-B expression could re-sensitize resistant cells to cisplatin (**C**). *: *p* < 0.05.

**Figure 5 cancers-13-05578-f005:**
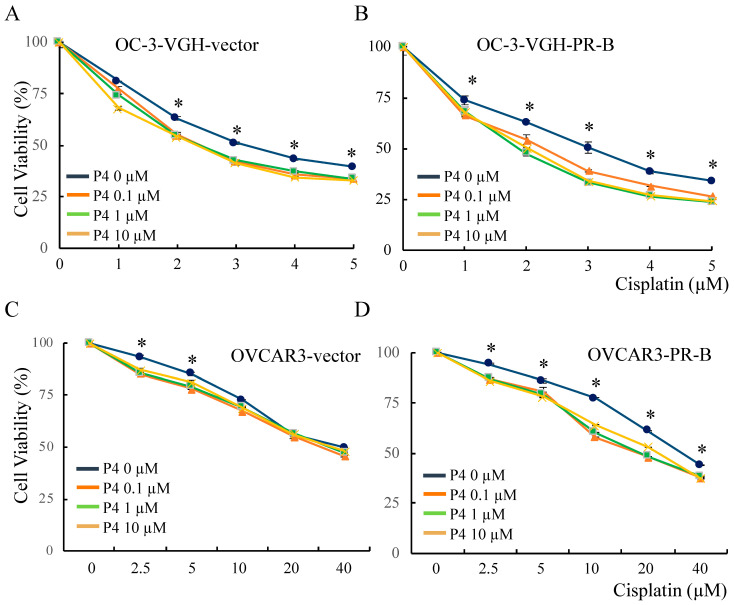
Progesterone (P4) increased platinum sensitivity of ovarian HGSC cells regardless of PR-B expression status. OC-3-VGH-vector (**A**), OC-3-VGH-PR-B (**B**), OVCAR-3-vector (**C**), and OVCAR-3-PR-B (**D**) cells were incubated with various concentrations of cisplatin for 24 h with different concentrations of P4. P4 promotes cell inhibition of cisplatin in all types of ovarian HGSC cells for about 10%. Cell viability was assessed by cell counting kit-8. *: *p* < 0.05.

**Figure 6 cancers-13-05578-f006:**
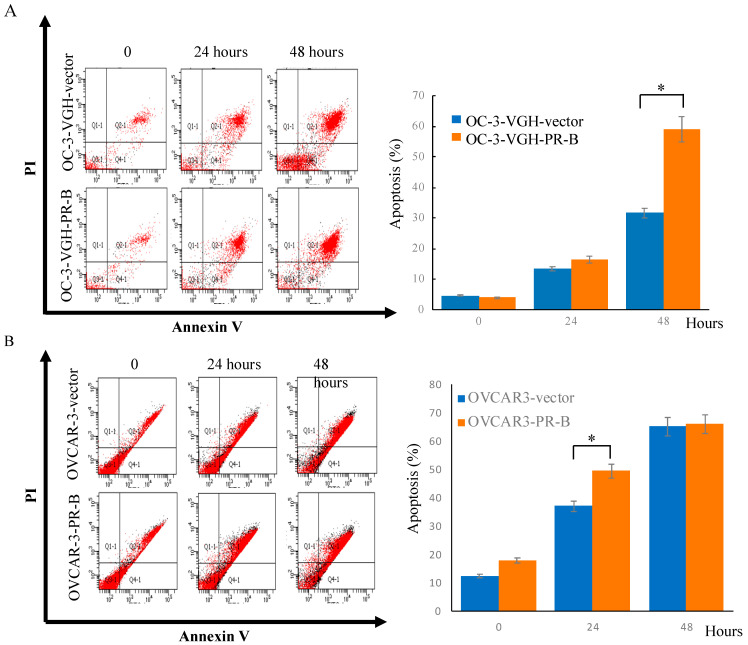
PR-B expression increased cisplatin related apoptosis of ovarian HGSC cells. Apoptosis assay using flow cytometry analysis showed that OC-3-VGH-PR-B (**A**) and OVCAR-3-PR-B (**B**) cells underwent significantly more apoptosis than OC-3-VGH-vector and OVCAR-3-vector cells after treatment of cisplatin for 48 and 24 h, respectively. *: *p* < 0.05.

**Table 1 cancers-13-05578-t001:** Demographic characteristics of patients according to platinum-sensitivity.

Parameter	Platinum Sensitive (*n* = 49)	Platinum Resistant (*n* = 41)	*p*-Value
Age (years), mean (SD)	55.6 (10.4)	56.4 (11.2)	0.731
CA-125 (U/mL), mean (SD)	1708.3 (2678.9)	2349.4 (3357.4)	0.093
CEA (ng/mL), mean (SD)	2.5 (4.3)	1.6 (1.3)	0.271
FIGO stage			
I	7 (14.3%)	0 (0%)	
II	6 (12.2%)	3 (7.3%)	
III	32 (65.3%)	28 (68.3%)	
IV	4 (8.2%)	10 (24.4%)	0.017
Optimal debulking	36 (73.5%)	12 (29.3%)	<0.001

Abbreviations: CA-125: carbohydrate antigen-125, CEA: carcinoembryonic antigen, FIGO: International Federation of Gynecology and Obstetrics, PR-B: progesterone receptor-B, SD: standard deviation, NACT: neo-adjuvant chemotherapy.

**Table 2 cancers-13-05578-t002:** Univariate and multivariate logistic regression analysis of the risk factors for predicting platinum sensitivity.

	Univariate Analysis			Multivariate Analysis		
Variable	OR	95% C.I.	*p* Value	OR	95% C.I.	*p* Value
Age ≥ 60	1.31	0.54–3.15	0.550			
Menopause	1.48	0.62–3.55	0.373			
Stage (III, IV)	4.57	1.20–17.4	0.026	2.83	0.66–12.2	0.163
Sub-optimal debulking	6.69	2.66–16.9	<0.001	5.56	2.09–14.8	0.001
CA125 ≥ 500 U/mL	2.27	0.97–5.29	0.059			
PR-B H-score < 12.5	4.02	1.43–11.3	0.008	3.69	1.18–11.6	0.025

Abbreviations: CA-125: carbohydrate antigen-125, C.I.: confidence interval, OR: odds ratio, PR-B: progesterone receptor-B.

## Data Availability

The data presented in this study are available on reasonable request from the corresponding author.

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
