# Peer review of "Highly Expressed Progesterone Receptor B Isoform Increases Platinum Sensitivity and Survival of Ovarian High-Grade Serous Carcinoma"

_cancers, 2021, doi:10.3390/cancers13215578_

Round 1
Reviewer 1 Report
The authors investigated the expression of progesterone-receptor (PR) in tissue from 90 patients of whom 41 recurred within 6 months after primary surgery and at least 4 courses with chemotherapy (platinum resistant group) and 49 patients with later relapse (platinum sensitive). PR expression was evaluated as total PR and as PR-B. Tissue from platinum sensitive patients had higher immunoscore for PR-B than tissue from platinum resistant patients. A high PR-B immunoscore indicated better survival. In multivariat regression analysis, high immunoscore and optimal debulking were the only independent factors for survival. Further, the authors performed lab studies on two cell lines (OVCAR-3) and (OC-3-VGH). In short; these studies showed that transfection of these 2 cell lines with PR-B made the cell lines more sensitive to cisplatin. Further, they established an acquired cisplatin resistant cell line from OC-3-VGH cells. When these cells were grown in a medium containing either cisplatin alone or a combination of cisplatin and progesterone. This experiment showed that the addition of progesterone increased the sensibility to cisplatin.
Evaluation:
This is an interesting and well performed study.
A minor comment:
The legend to Figure 3 contains an error concerning C.
“ OC-3-VGH-PR-B/OVCAR-3-PR-B cells express higher PR-B both in protein and mRNA levels, and are more Scheme 3. VGH-vector/OVCAR-3-vector cells. *: P<0.05”
This sentence does not make sense.
Author Response
Comments and Suggestions for Authors
The authors investigated the expression of progesterone-receptor (PR) in tissue from 90 patients of whom 41 recurred within 6 months after primary surgery and at least 4 courses with chemotherapy (platinum resistant group) and 49 patients with later relapse (platinum sensitive). PR expression was evaluated as total PR and as PR-B. Tissue from platinum sensitive patients had higher immunoscore for PR-B than tissue from platinum resistant patients. A high PR-B immunoscore indicated better survival. In multivariat regression analysis, high immunoscore and optimal debulking were the only independent factors for survival. Further, the authors performed lab studies on two cell lines (OVCAR-3) and (OC-3-VGH). In short; these studies showed that transfection of these 2 cell lines with PR-B made the cell lines more sensitive to cisplatin. Further, they established an acquired cisplatin resistant cell line from OC-3-VGH cells. When these cells were grown in a medium containing either cisplatin alone or a combination of cisplatin and progesterone. This experiment showed that the addition of progesterone increased the sensibility to cisplatin.
Evaluation:
This is an interesting and well performed study.
A minor comment:
The legend to Figure 3 contains an error concerning C.
“ OC-3-VGH-PR-B/OVCAR-3-PR-B cells express higher PR-B both in protein and mRNA levels, and are more Scheme 3. VGH-vector/OVCAR-3-vector cells. *: P<0.05”
This sentence does not make sense.
= Thank you for your comment. We have corrected the legend to Figure 3. We changed the sentence to “OC-3-VGH-PR-B/OVCAR-3-PR-B cells show higher PR-B expression at both the protein and mRNA levels and are more sensitive to cisplatin treatment than OC-3-VGH-vector/OVCAR-3-vector cells. *: P<0.05.”, on line-329 to -332.

Reviewer 2 Report
In the submitted manuscript Lin at al. have show that in high-grade serous ovarian cancer (HGSOC) patients a higher expression of progesterone receptor B is a marker for better prognosis and indicator of higher sensitivity to platinum-based therapy. On in vitro level they showed that overexpression PR-B or progesterone treatment could increase cisplatin-sensitivity in HGSOC cell by promoting the cisplatin-related apoptosis.
Although somehow interesting and potentially translatable to the clinics, this manuscript has multiple both major and minor drawbacks:
1) First and foremost the quality of English language is very poor, lots of words are missing, adverbs were used instead of adjectives, wrong tenses were used, etc. Therefore, this manuscript must undergo a professional English proofreading.
2) Line 81: It cannot be said that "PR expression improved survival of ovarian HGSC", PR expression could only be related to or associated with better survival.
3) Line 103: Provide the reference for FIGO 2014 "system".
4) Lines 109-110: Provide a reference for the method you used to classify patients to platinum-sensitive and -resistant groups.
5) Provide dilutions for ALL used primary antibodies (lines 120-121 and 158).
6) Explain in much more details how did you created cisplatin-resistant OC-3-VGH cell line because your description is irreproducible.
7) Provide type and vendor of imaging system used for blots. Explain how western blots were quantified.
8) Line 168: What does sentence "Prism software was used to analyze results." mean here? I believe that you used a software for Applied Biosystems 7500 and that used SPSS software package for statistical analyses.
9) If you use https://mfeprimer3.igenetech.com/spec tool you could see that your PR-B primers is highly unspecific! Provide an agarose gel of qPCR amplicons to prove that your qPCR multiplies only PR-B isoform.
10) 'Statistical analysis' subsection must be rewritten because it is confusing. First you wrote "For comparisons of median and mean values, two-sample t-test was performed for data with normal distribution and non-parametric test was applied to analyze data without normal distribution." without specifying which particular non-parametric test was used, and than you wrote "The Student’s t-test or one-way ANOVA was used to analyze statistical significance between control and treatment groups (RT-PCR and western blotting). The Bonferroni pair test was used to assess differences between control and experimental groups (CCK-8 assay)." what if totally confusing. In addition, what is "The Bonferroni pair test"?! I suppose you meant paired t-test with Bonferroni correction, but why was only for CCK-8 paired test and Bonferroni correction used? Describe what data is presented on graphs (mean+-SD or ...).
10) Line 218 and throughout the text: I have a problem with denoting a PR-B expression with a H-score higher that 12.5 as "strong", since maximal H-score could be 300. This is misleading since cut-off was based on a platinum-sensitivity, not extent of PR-B expression!
11) Lines 230-231: Log-rang test compares the whole survival curves, not just 5-year survival rates. So here you can either provide actual 5-year survival curves or hazard ratios, not just P-values.
12) Figure 1 legend is a mess. From it I cannot really understand what Figure 1 presents. Representative IHC images should be provided for all levels of PR expression, for both PR and PR-B. For scatter plot also provide Pearson's correlation coefficient and its P-values. In addition, a correlation graph would look better with a logarithmic scale. Significance lines should be placed over the center of bars.
13) On Y-axes of survival curves, survival rates should be placed next to the tickmarks. X-axes should be divided by more intuitive increments, like 24 months (2 years).
14) In tables there is no point to mark P-values less than 0.05 with an asterisk whose explanation is "p<0.05". Everyone understands which numbers are <0.05!
15) Table 2. Multivariate analysis is usually conducted and presented with only those variables which showed P<0.05 in univariate analysis. Similarly, you should conduct a Cox proportional hazards regression to see if PR-B expression is actually an independent biomarker for PFS and DSS!
16) Sentence in lines 254-256 is unclear because it seems that OC-3-VGH and OVCAR-3 cell were transfected with a different control and PR-B vector. IMHO, this sentence should succeed the next one and actually explain which all types of cell lines were created.
17) As noted previously, explain how immunoblotts were actually quantified.
18) Figure 3 legend is unclear/wrong: "(B) (C). OC-3-VGH-PR-B/OVCAR-3-PR-B cells express higher PR-B both in protein and mRNA levels, and are more Scheme 3. VGH-vector/OVCAR-3-vector cells.".
19) Explain what is P4, and provide vendor of progesterone in 'Materials and Methods'.
20) Lines 325-326: Recheck all references because [23] is not Lee at el.!
21) Line 342: Recheck names of the cited authors because first authors of reference [25] is Peluso, not Preluso.
22) Line 379: If you claim that anti-PR-B antibody is a PR-B specific then you can easily show there was a constant ratio of PR-A to PR-B in your patients! There is a simple formula for calculating PR-A expression: PR H-score minus PR-B H-score. Much more emphasis should be put on the actual or potential significance of PR (or PR-A)/PR-B ratio.
23) Line 382: What is "the B upstream segment"?
24) This manuscript generally needs more data on the structure of PGR gene and the significance and different roles of its isoforms. Even further, if you look at NCBI and UniProt databases you will see that it has more transcripts/databases than just B and A!
Author Response
Reviewer 2
Comments and Suggestions for Authors
In the submitted manuscript Lin at al. have show that in high-grade serous ovarian cancer (HGSOC) patients a higher expression of progesterone receptor B is a marker for better prognosis and indicator of higher sensitivity to platinum-based therapy. On in vitro level they showed that overexpression PR-B or progesterone treatment could increase cisplatin-sensitivity in HGSOC cell by promoting the cisplatin-related apoptosis.
Although somehow interesting and potentially translatable to the clinics, this manuscript has multiple both major and minor drawbacks:
1) First and foremost the quality of English language is very poor, lots of words are missing, adverbs were used instead of adjectives, wrong tenses were used, etc. Therefore, this manuscript must undergo a professional English proofreading.
= Thank you for your comment. We have sent our manuscript to the English editing service of MDPI to improve the quality of English.
2) Line 81: It cannot be said that "PR expression improved survival of ovarian HGSC", PR expression could only be related to or associated with better survival.
= Thank you for your comment. We changed the sentence to “Additionally, PR expression could be associated with better survival of ovarian HGSC”, on line-105 ton -106.
3) Line 103: Provide the reference for FIGO 2014 "system".
= Thank you for your comment. We added “Mutch, D. G.; Prat, J., 2014 FIGO staging for ovarian, fallopian tube and peritoneal cancer. Gynecol Oncol 2014, 133, (3), 401-4.” as the reference for FIGO 2014 system, on line-128, as the 20th referece.
4) Lines 109-110: Provide a reference for the method you used to classify patients to platinum-sensitive and -resistant groups.
= Thank you for your comment. We have provided a reference for the definition of platinum-sensitivity (J Clin Oncol 1991, 9, (3), 389-93).”, as the 21th reference, on line-136.
5) Provide dilutions for ALL used primary antibodies (lines 120-121 and 158).
= Thank you for your comment. We diluted the antibodies: Leica, US. Cat# PR NCL-L-PGR-312, 1:100 and Merck, USA.,Cat# MABS1234 , 1:100, as the primary antibodies for immunohistochemical staining, on line-149 and -150.
6) Explain in much more details how did you created cisplatin-resistant OC-3-VGH cell line because your description is irreproducible.
= Thank you for your comment. We changed the sentences to “We developed acquired platinum-resistant OV3-VGH cells using a stepwise increase in treatment concentrations with cisplatin (Sigma, St. Louis, USA, Cat# 479306) (1 to 6 μM). This development period was carried out for about 2 months. Finally, the cisplatin-resistant cells (OC-3-VGH-resist) were kept in the DMEM/F12 medium with 6 μm cisplatin.”, on line -170 to 175.
7) Provide type and vendor of imaging system used for blots. Explain how western blots were quantified.
= Thank you for your comment. We have added “The images were read by a LAS-3000 imager system (Fujifilm). Densitometric quantification of the bends was performed using Bio-Rad Quantity One 1-D Analysis software. Relative levels of PR-B expression were determined by normalization to the expression of β-actin.”, on line-195 to -198.
8) Line 168: What does sentence "Prism software was used to analyze results." mean here? I believe that you used a software for Applied Biosystems 7500 and that used SPSS software package for statistical analyses.
= Thank you for your comment. We do use Applied Biosystems 7500 system to read the results of RT-PCR and analyze the data with SPSS software package. So, we have changed the sentence “Prism software was used to analyze results.” to “SPSS software was used to analyze results.”, on line-205 to -206.
9) If you use https://mfeprimer3.igenetech.com/spec tool you could see that your PR-B primers is highly unspecific! Provide an agarose gel of qPCR amplicons to prove that your qPCR multiplies only PR-B isoform.
= Thank you for your comment. The PR-B primers we used in present study are highly specific when you look at the information provided at the website in detail. An agarose gel of qPCR amplicons is provided as below:
The amplification of PR-B was performed with an initial denaturing, 95-C for 5 min; 40 cycles of repeated denaturing at 94-C for 30 s; annealing at 58-C for 50 s; and extension at 72-C for 30 s.
10) 'Statistical analysis' subsection must be rewritten because it is confusing. First you wrote "For comparisons of median and mean values, two-sample t-test was performed for data with normal distribution and non-parametric test was applied to analyze data without normal distribution." without specifying which particular non-parametric test was used, and then you wrote "The Student’s t-test or one-way ANOVA was used to analyze statistical significance between control and treatment groups (RT-PCR and western blotting). The Bonferroni pair test was used to assess differences between control and experimental groups (CCK-8 assay)." what if totally confusing. In addition, what is "The Bonferroni pair test"?! I suppose you meant paired t-test with Bonferroni correction, but why was only for CCK-8 paired test and Bonferroni correction used? Describe what data is presented on graphs (mean+-SD or ...).
= Thank you for your comment. We re-wrote the paragraph, on line-231 to -252.
For comparisons of mean values, student t-test was used for “age” and non-parametric test was used for “CEA, CA-125, treat-free interval and PR-B score”, according to normal distribution or not. We used two-way ANOVA with Bonferroni post hoc test to assess the differences of viability between control and experimental groups (CCK-8 assay) treated with cisplatin.
We have changed the presentation of data on graphs with mean ± SD, on line-269, -312, -320 to -321 and -324.
10) Line 218 and throughout the text: I have a problem with denoting a PR-B expression with a H-score higher that 12.5 as "strong", since maximal H-score could be 300. This is misleading since cut-off was based on a platinum-sensitivity, not extent of PR-B expression!
= Thank you for your comment. According to the characteristics of patients, we found the level of PR-B H-score was significantly higher in platinum-sensitive group. We tried to find the cut-off level of PR-B H-score to predict platinum sensitivity. So, we used ROC curves and Youden index to find the optimal cut-off level of PR-B H-score.
11) Lines 230-231: Log-rang test compares the whole survival curves, not just 5-year survival rates. So here you can either provide actual 5-year survival curves or hazard ratios, not just P-values.
= Thank you for your comment. We have added supplementary Table 1 and 2 showing hazard ratios of PFS and CSS. Survival curves were shown in Figure 2A and 2B.
12) Figure 1 legend is a mess. From it I cannot really understand what Figure 1 presents. Representative IHC images should be provided for all levels of PR expression, for both PR and PR-B. For scatter plot also provide Pearson's correlation coefficient and its P-values. In addition, a correlation graph would look better with a logarithmic scale. Significance lines should be placed over the center of bars.
=Thanks for your comment. We have changed the legend to “(A, B, C) Representative images for PR-B immunohistochemical staining (scale bar=200μm). The intensities were strong (A), moderate (B) and weak(C). The percentages of positive cells were 70% (A), 20% (B) and 3% (C). H-scores of (A), (B) and (C) were 210, 40 and 3, respectively. (D) The scatter plot (log-log scale) shows the positive correlation between total PR and PR-B H-score; Pearson’s correlation coefficient was 0.831, p<0.001. (E) Data were expressed as mean¬¬¬ PR H-score; platinum-sensitive patients had significantly higher H-score than platinum-resistant patients, p=0.005.”, on line-285 to 293. And, we have changed figure 1D with a logarithmic scale.
13) On Y-axes of survival curves, survival rates should be placed next to the tickmarks. X-axes should be divided by more intuitive increments, like 24 months (2 years).
=Thanks for your comment. We have changed the Figure 2.
14) In tables there is no point to mark P-values less than 0.05 with an asterisk whose explanation is "p<0.05". Everyone understands which numbers are <0.05!
=Thanks for your comment. We have deleted the asterisk in tables.
15) Table 2. Multivariate analysis is usually conducted and presented with only those variables which showed P<0.05 in univariate analysis. Similarly, you should conduct a Cox proportional hazards regression to see if PR-B expression is actually an independent biomarker for PFS and DSS!
=Thanks for your comment.
We have made an amendment changes on Table 2 with multivariate analysis only include those variables which showed p<0.05 in univariate analysis.
We have done a Cox proportional hazards regression analysis for PFS and CSS of our cohort. Data were shown on supplemental Table 1 and Table 2. We have added “In multivariate Cox regression analysis, optimal debulking status was the only significant independent factor associated with both PFS and CSS, while PR-B was marginal significance only for PFS (supplementary Table 1, 2).”, on line-280 to -282.
16) Sentence in lines 254-256 is unclear because it seems that OC-3-VGH and OVCAR-3 cell were transfected with a different control and PR-B vector. IMHO, this sentence should succeed the next one and actually explain which all types of cell lines were created.
=Thanks for your comment. We transfected the cells (OC-3-VGH and OVCAR-3 cell) with pcDNA3-PR-B and pcDNA3 (as vector). We selected the stable clones with G418. After selection, we had four stable clones: OC-3-VGH-PR-B, OC-3-VGH-vector, OVCAR-3-PR-B and OVCAR-3-vector. The data were shown in Material and Methods 2.4.
17) As noted previously, explain how immunoblotts were actually quantified.
=Thanks for your comment. The images were read by a LAS-3000 imager system (Fujifilm). Densitometric quantification of the bends was performed using Bio-Rad Quantity One 1-D Analysis software. Relative levels of PR-B expression were determined by normalization to the expression of β-actin, on line-195 to -198.
18) Figure 3 legend is unclear/wrong: "(B) (C). OC-3-VGH-PR-B/OVCAR-3-PR-B cells express higher PR-B both in protein and mRNA levels, and are more Scheme 3. VGH-vector/OVCAR-3-vector cells.".
= Thanks for your comment. We have changed the sentence to “(B) (C). OC-3-VGH-PR-B/OVCAR-3-PR-B cells show higher PR-B expression at both the protein and mRNA levels and are more sensitive to cisplatin treatment than OC-3-VGH-vector/OVCAR-3-vector cells. *: P<0.05.”, on line-329 to -332.
19) Explain what is P4, and provide vendor of progesterone in 'Materials and Methods'.
= Thanks for your comment. We add the sentence “ progesterone, (P4, TargetMol, #T0478)” in Material and Methods 2.7., on line-217 to -218.
20) Lines 325-326: Recheck all references because [23] is not Lee at el.!
= Thanks for your comment. We have rechecked all the references and corrected the error. The report was cited from Tkalia et al. and the number of references is 25, on line-401 to 402.
21) Line 342: Recheck names of the cited authors because first authors of reference [25] is Peluso, not Preluso.
= Thanks for your comment. We have changed Preluso to Peluso, the number of the reference is 27, on line-419 to 421.
22) Line 379: If you claim that anti-PR-B antibody is a PR-B specific then you can easily show there was a constant ratio of PR-A to PR-B in your patients! There is a simple formula for calculating PR-A expression: PR H-score minus PR-B H-score. Much more emphasis should be put on the actual or potential significance of PR (or PR-A)/PR-B ratio.
= Thanks for your comment. We agree with your opinion and the formula might be feasible. However, a confirmatory study using western blotting to discriminate between the isoforms is still needed before getting to a conclusion.
23) Line 382: What is "the B upstream segment"?
= Thanks for your comment. We have changed the sentence to “Some antibody clones recognize epitope on the N-terminal of PR-B, so they specifically detect PR-B only.”, on line -451 to -452.
24) This manuscript generally needs more data on the structure of PGR gene and the significance and different roles of its isoforms. Even further, if you look at NCBI and UniProt databases you will see that it has more transcripts/databases than just B and A!
= Thanks for your comment. In this study, we focused to investigate the association between PR expression, PR isoform B, chemosensitivity, and survival in ovarian HGSC patients and cell-line models. We have added the sentence “only PR-B and total PR were investigated” on line-460, as one of the limitations of the study. We will conduct further study to investigate PR-A, PR-B, PR-C, total PR, and PR membranous component-1 of ovarian cancer.

Reviewer 3 Report
61-62: new data with PARP inhibitors show improved DFS
74: I think it is a strong statement to say that it shows the role of progesterone in cancer development.
212: I would be interested to know why so many non HGSC were excluded (majority of cases) since it is the most common sub-type of ovarian cancer.
213: Did all patients need to have primary debulking surgery followed by adjuvant chemo or could they have NACT with interval debulking?
223: than those with platinum-resistant disease
242: I don't understand the TFI in the table. These are patients with primary diagnosis of ovarian cancer and not relapsed. Does it represent time to next treatment if recurrence? If so, I would not put it in table 1 and it would need to be bette described. I would also removed the section where stages are presented as early or advanced. Also, big difference in optimal debulking between both groups. Hence my previous question to know whether some patients were offered NACT (which would allow for more optimal debulking in some patients). PR-B-H score should not be in table 1.
There is not data on the genetic aspect of these patients. Up to 25% will have either germline or somatic BRCA mutations that are associated with increased platinum-sensitivity.
364: I would be careful comparing HGS to LGS ovarian cancer because underlying history, genetic mutations and management are very different.
399: I don't think the data is there to say that this approach has the potential to be used clinically to improve efficacy of cisplatin. The only conclusion is that further research is required. But I also don't know how much can be extrapolated because cisplatin is not used (or at least is not the standard of care) for management of HGS EOC. It is rather carboplatin that is used in conjunction with placlitaxel.
Author Response
Review 3
Comments and Suggestions for Authors
61-62: new data with PARP inhibitors show improved DFS
= Thanks for your comment. We agree PARP inhibitor showed improved DFS. We have deleted the sentence “and the overall …..has not changed substantially over the past 2 decades.”, on line-82.
74: I think it is a strong statement to say that it shows the role of progesterone in cancer development.
= Thanks for your comment. We have changed “indicating” to “implying”, on line-96 to -97.
212: I would be interested to know why so many non HGSC were excluded (majority of cases) since it is the most common sub-type of ovarian cancer.
= Thanks for your comment. According the Taiwan's Nationwide Cancer Registry Annual Report 2018, available at: https://www.hpa.gov.tw/Pages/ashx/File.ashx?FilePath=~/File/Attach/13498/File_15611.pdf, accessed October 25, 2021, serous carcinoma the most common form of ovarian cancer and accounts for only 31.73% of all ovarian cancer cases in Taiwan. Followed by clear cell carcinoma, endometrioid carcinoma and mucinous carcinoma, which account for 18.50%, 14.96% and 9.63%, respectively. In our cohort, serous carcinoma accounts for 34.3% (129/376). The ratio is similar to nationwide data of Taiwan.
213: Did all patients need to have primary debulking surgery followed by adjuvant chemo or could they have NACT with interval debulking?
= Thanks for your comment. According our inclusion criteria, we included patients with primary (77 patients) and interval debulking surgery (13 patients).
223: than those with platinum-resistant disease
= Thanks for your comment. We have changed “patients” to “disease”, on line-268 -269.
242: I don't understand the TFI in the table. These are patients with primary diagnosis of ovarian cancer and not relapsed. Does it represent time to next treatment if recurrence? If so, I would not put it in table 1 and it would need to be better described. I would also removed the section where stages are presented as early or advanced. Also, big difference in optimal debulking between both groups. Hence my previous question to know whether some patients were offered NACT (which would allow for more optimal debulking in some patients). PR-B-H score should not be in table 1.
= Thanks for your comment. We have removed TFI, low/high stage and PR-B-H score from table 1. There were 13 patients received NACT, 9 patients (69.2%) achieved optimal debulking. The remaining 77 patients received primary debulking surgery and 39 of them (50.9%) achieved optimal debulking. We found the trend that patients with NACT had higher possibility of optimal debulking, but it did not reach statistical significance, p=0.214.
There is not data on the genetic aspect of these patients. Up to 25% will have either germline or somatic BRCA mutations that are associated with increased platinum-sensitivity.
= Thanks for your comment. We agree that patients with BRCA mutations associated with increased platinum-sensitivity. Due to inadequate research funding, we did not have the BRCA mutations data from all of our patients.
364: I would be careful comparing HGS to LGS ovarian cancer because underlying history, genetic mutations and management are very different.
= Thanks for your comment. We have removed the sentences about LGS, in the paragraph between line-446 to -447.
399: I don't think the data is there to say that this approach has the potential to be used clinically to improve efficacy of cisplatin. The only conclusion is that further research is required. But I also don't know how much can be extrapolated because cisplatin is not used (or at least is not the standard of care) for management of HGS EOC. It is rather carboplatin that is used in conjunction with placlitaxel.
= Thanks for your comment. Aabo et al*. systemically reviewed the data from 37 trial, 5667 patients. In the paragraph of “Carboplatin versus cisplatin” in this article, they did not find good evidence of any difference in efficacy between cisplatin and carboplatin.
We will conduct the further study with this issue.
*British Journal of Cancer (1998) 78(11), 14i79-1487

Round 2
Reviewer 2 Report
Authors have satisfactorily responded to all my questions and concerns, and substantially improved the quality of this manuscript.